# MacLaSa: Multi-Aspect Controllable Text Generation via Efficient Sampling from Compact Latent Space

**Hanxing Ding**[1,2]   **Liang Pang**[1]*   **Zihao Wei**[1,2]   **Huawei Shen**[1,2]
**Xueqi Cheng**[1,2]   **Tat-Seng Chua**[3]

[1]Institute of Computing Technology, Chinese Academy of Sciences
[2] University of Chinese Academy of Sciences
[3]Sea-NExT Joint Lab, National University of Singapore
{dinghanxing18s, pangliang, weizihao22z, shenhuawei, cxq}@ict.ac.cn
dcscts@nus.edu.sg

## Abstract

Multi-aspect controllable text generation aims to generate fluent sentences that possess multiple desired attributes simultaneously. Traditional methods either require expensive iteration / searching within the discrete text space during the decoding stage, or train separate controllers for each aspect, resulting in a degradation of text quality due to the discrepancy between different aspects. To address these limitations, we introduce a novel approach for **M**ulti-**a**spect **c**ontrol, namely MacLaSa, that estimates compact **La**tent space for multiple aspects, and performs efficient **Sa**mpling with a fast sampler. To eliminate the domain discrepancies between different aspects, we first utilize a variational autoencoder (VAE) network to map text sequences from various data sources into close latent representations. The estimated latent space enables the formulation of joint energy-based models and the plugging in of arbitrary attribute discriminators to achieve multi-aspect control. Afterwards, we draw latent samples with a fast sampler based on ordinary differential equations and feed sampled examples to the VAE decoder to produce target text sequences. Experimental results demonstrate that MacLaSa outperforms strong baselines on both attribute relevance and textual quality while maintaining a high inference speed.

## 1 Introduction

Attribute-based controllable generation aims to generate text that exhibits desired attributes in certain aspects (Zhang et al., 2022). Early work focused on single-aspect control tasks and involved re-training or fine-tuning language models (LMs) using well-labeled data, which resulted in good performance (Keskar et al., 2019; Chan et al., 2021; Hao et al., 2021; Hu et al., 2017; Ficler and Goldberg, 2017; Xiao et al., 2021). Recent studies focus on a more challenging and practical setting,

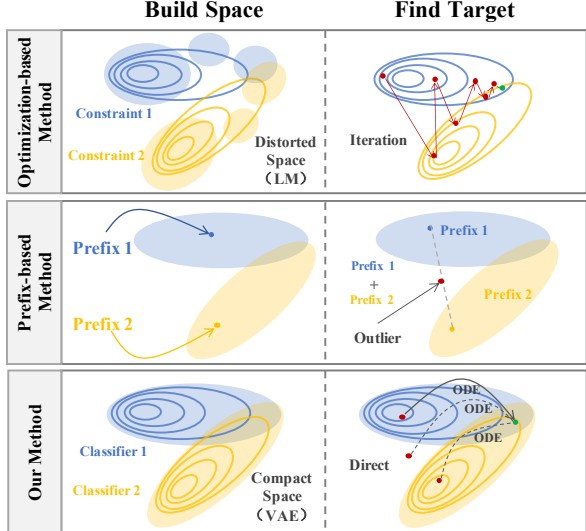

Figure 1: A comparison of existing methods and our approach. **Top:** optimization-based methods perform iteration / searching in the distorted text space. **Middle:** prefix-based methods fuse multiple prefixes to obtain interpolation or average of these distribution centers. **Bottom:** our framework estimates compact latent space for better controllability and performs efficient sampling with a fast ODE-based sampler.

multi-aspect controllable text generation[1] (Kumar et al., 2021; Qian et al., 2022; Qin et al., 2022). For instance, a dialogue system may require the control of emotions, persona, politeness, etc, at the same time. However, training multi-aspect controllers directly is difficult due to the limited availability of sentences with multi-attribute annotations. Thus, recent works focus on training separate single-aspect discriminators or controllers for each aspect and combining them for multi-aspect controllable text generation (Mireshghallah et al., 2022; Qian et al., 2022).

As illustrated in Figure 1, recent works on multi-aspect controllable text generation task can be primarily categorized into two types. Firstly,

---

*Corresponding author

[1]The aspect can be sentiment or topic, and sentiment may have two attributes: positive and negative.

optimization-based methods either apply extra attribute classifiers to adjust the conditional probability distributions of language model at every generation step (Dathathri et al., 2020; Krause et al., 2021; Yang and Klein, 2021), or regard the decoding process as an optimization objective and search for optimal soft-representations that satisfy multi-objective constraints (Kumar et al., 2021, 2022; Qin et al., 2022; Mireshghallah et al., 2022). However, from a distributional perspective, optimization-based methods often conduct complicated gradient-descent iterations or searching in the distorted text space, and the discrete nature makes it difficult to find high-quality texts, leading to poor linguistic quality and slow inference speeds. Secondly, prefix-based methods are introduced to guide conditional generation using lightweight continuous task-specific vectors (Qian et al., 2022; Yang et al., 2022). They typically train single-aspect prefixes separately and suffer from text quality degeneration when combining them for multi-aspect control due to the mutual interference between multiple prefixes. As depicted in Figure 1, prefix-based methods combine multiple prefixes to obtain the interpolation or average of these distribution centers appraised by prefixes. However, there could be a mismatch between interpolation points and target intersection regions when the distribution centers of different aspects are far away, leading to the degradation of textual fluency. Therefore, an ideal method for multi-aspect controllable generation should enhance controllability and textual quality, while enabling rapid inference speeds.

In this paper, we introduce a new technique for multi-aspect controllable text generation, dubbed MacLaSa, which estimates a compact space containing latent representations of various attributes and performs effective sampling using a fast sampler based on ordinary differential equations (ODEs). To eliminate the domain discrepancies between different aspects, we initially employ a VAE encoder network to map attribute-related sentences into latent representations and penalize the distance between each pair of aspect distribution centers. The acquired compact latent space aids in formulating joint latent-space energy-based models (EBMs) and allows us to integrate arbitrary attribute discriminators to satisfy multi-aspect combinations. Subsequently, we utilize an efficient ODE-based sampler (Song et al., 2021; Nie et al., 2021) to draw latent samples possessing desired attributes

from the distribution formed by multiple attribute classifiers. Ultimately, the selected latent vectors are input into a VAE decoder to generate target text sequences. In short, our approach improves controllability and textual quality by estimating a compact latent space to mitigate mutual interference among various aspects, and the fast ODE-based sampler contributes to efficient sampling.

We conduct experiments on the multi-aspect control task with two attributes from the sentiment aspect and four attributes from the topic aspect, with datasets IMDb movie reviews (Maas et al., 2011) and AGNews (Zhang et al., 2015), respectively. Experimental results of both automatic and human evaluation demonstrate that our method achieves encouraging improvements in attribute relevance and text quality compared to previous strong baselines. Our work also exhibits significant advantages in inference speed over existing baselines[2].

## 2 Related Work

In this section, we discuss the related work on multi-aspect control. Recent researches on multi-aspect can be divided into two types: optimization-based methods and prefix-based methods.

**Optimization-based Methods** Existing efforts on multi-aspect control typically combine many attribute controllers in the decoding stage to bias the language model for desired directions. Weighted-decoding methods focus on decomposing conditional probability through Bayesian factorization into a language model and a classifier (Dathathri et al., 2020; Krause et al., 2021; Yang and Klein, 2021; Liu et al., 2021; Gu et al., 2022a; Hallinan et al., 2023). Other approaches define controllable text generation as a multi-objective optimization problem and find the optimal soft-representation sequences by specific sampling schemes or other gradient-based samplers (Lample et al., 2018; Bhattacharyya et al., 2021; Mireshghallah et al., 2022; Qin et al., 2022; Kumar et al., 2021, 2022). These optimization-based methods often require complicated iteration / search in the high-dimensional text space, leading to slow inference speed.

**Prefix-based Methods** Recent work leverages the learned continuous task-specific vectors, which are called prefixes, as a lightweight alternative to guide the language model to generate desired

[2]Our code is available at https://github.com/TrustedLLM/MacLaSa

attribute text (Li and Liang, 2021; Yu et al., 2021; Zhao et al., 2020; Qian et al., 2022; Yang et al., 2022; Huang et al., 2023). Contrastive Prefixes (Qian et al., 2022) utilize the opposite relationship between different attributes to help to train single-aspect prefixes and combine them for multi-aspect control. Tailor (Yang et al., 2022) provides a multi-aspect prefix mask and a re-indexing position-ids sequence to bridge the gap between single and multi-aspect control. Nevertheless, these learned controllers in prefix-based methods may prefer different language habits, resulting in textual quality degeneration when combining them for multi-aspect control.

There is also a line of work that manipulates latent variables in the latent space (Gu et al., 2022c,b; Liu et al., 2022). Gu et al. (2022c) map attribute-related sentences to the latent space and then designs a heuristic searching algorithm to approach intersection regions of the different attributes for generation. Despite their efficiency, they still suffer from the unstable controllability due to the rare intersections of different attributes. LatentOps (Liu et al., 2022) executes composable control operations within the low-dimensional continuous latent space. However, it does not adequately consider the discrepancy between various aspects, resulting in suboptimal performance when controlling multiple attributes simultaneously.

## 3 Methodology

In this section, we first present the task definition of multi-aspect controllable text generation (§3.1). Next, we describe how to build the compact latent space (§3.2), how to define the joint EBMs on the latent space (§3.3), and how to sample from the EBMs to generate the final results (§3.4).

The overall structure of MacLaSa is illustrated in Figure 2. Our approach primarily relies on the variational autoencoder architecture for manipulating latent spaces. To weaken the mutual interference among different aspects, we initially employ the VAE encoder to estimate a continuous low-dimensional latent space, incorporating additional losses to ensure its compactness. Subsequently, we establish joint latent-space energy-based models, which allow us to integrate multiple constraint functions for guiding sophisticated multi-aspect control. Finally, we utilize a fast ODE-based sampler to draw samples from the EBMs and input them into the VAE decoder to generate the desired

multi-aspect sequences.

### 3.1 Task Definition

First, we present the task definition of multi-aspect controllable text generation. Suppose we have $N$ aspects, represented by $\mathbf{A} = \{A_1, \cdots, A_N\}$, where each aspect $A_n$ contains $|A_n|$ attributes, given by $\{a_n^1, \cdots, a_n^{|A_n|}\}$. The goal of multi-aspect control is to generate sentences that possess multiple attributes $\boldsymbol{a} = \{a_1^*, \cdots, a_N^*\}$ simultaneously. For instance, we may expect our model to produce a sentence with attribute $a_1^2$ (from aspect $A_1$) and attribute $a_2^4$ (from aspect $A_2$).

Our training samples are organized and labeled according to their corresponding aspects and attributes. $S_n^j$ denotes the index set of sentences with attribute $a_n^j$. As a result, we have $S_n = \bigcup_{j=1}^{|A_n|} S_n^j$, which represents the index set containing all sentences within aspect $A_n$. Likewise, $S = \bigcup_{n=1}^{N} S_n$ signifies the indices encompassing our entire training dataset. We use $x$ to represent an arbitrary sentence and $z$ to indicate its latent representation.

It is worth noting that our training corpus contains only single-aspect labeled sentences, making it infeasible to directly train a multi-aspect controllable text generative model.

### 3.2 Building Latent Space

To estimate a compact, continuous latent space that outlines the latent distribution of interest and facilitates subsequent sampling processes, we utilize a VAE network equipped with pre-trained language models to encode any single-aspect sentence $x$ to its hidden representation $z$ using $z = \text{Encoder}_\phi(x)$. The encoded latent representations constitute the estimated attribute space.

We expect the latent space to be sufficiently compact while ensuring that latent representations from various aspects maintain their semantic meanings. To accomplish this, we propose the following three training objectives:

**ELBO Loss $\mathcal{L}_E$** We adopt the basic Evidence Lower Bound (ELBO) objective to learn a smooth latent space and force the decoder to map any given latent vector $z$ into its original text $x$:

$$\mathcal{L}_{\mathrm{E}} = -\mathbb{E}_{q_\phi(z|x)}[\log p_\theta(x|z)] + \mathrm{KL}(q_\phi(z|x)\|p_{\mathrm{prior}}(z)),$$
(1)

where $p_{\mathrm{prior}}(z)$ is a standard Gaussian distribution as the prior, and $\mathrm{KL}(\cdot\|\cdot)$ is the Kullack-Leibler divergency. The first term encourages $z$ to encode more relevant content information for reconstruct-

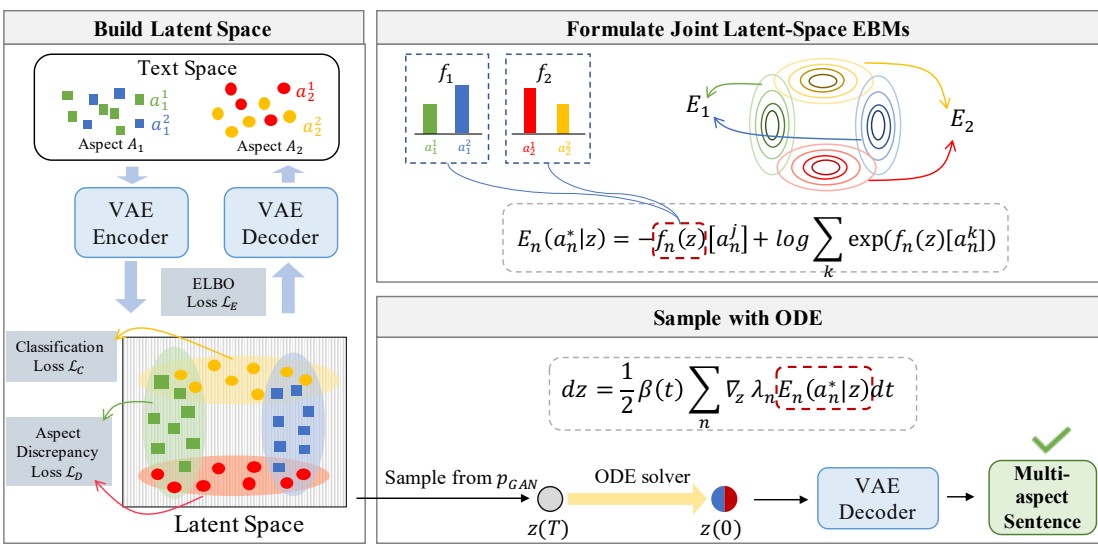

Figure 2: An overview of MacLaSa. **Left**: Build latent space for MacLaSa. We utilize the VAE framework with two additional losses to build a compact latent space. **Top Right**: Formulate joint EBMs. We formulate the latent-space EBMs of latent representation and attribute to facilitate the plug in of multiple attribute constraint classifiers. **Bottom Right** Sample with ODE. We adopt a fast ODE-based sampler to perform efficient sampling from the EBMs, and feed samples to the VAE decoder to output desired multi-aspect sentences.

ing the original text $x$ with the VAE decoder $p_\theta$. The KL divergence forces the variational distribution $q_\phi(z|x)$ to match the prior.

**Classification Loss $\mathcal{L}_C$** We propose the classification loss $\mathcal{L}_C$ to force the mapped representations to preserve their original attribute information and help the model to distinguish representations of different attributes from the same aspect. We introduce independent classification layers for each aspect and train them by minimizing the negative log-likelihood of the corresponding attribute $a_n^j$:

$$\mathcal{L}_C = -\sum_{n=1}^{N} \sum_{j=1}^{|A_n|} \sum_{i \in S_n^j} \log p_{\pi_n}\left(a_n^j \mid z_i\right), \quad (2)$$

where $p_{\pi_n}$ is a classifier that distinguish attributes $\{a_n^*\}$ from aspect $A_n$ with parameter $\pi_n$.

**Aspect Discrepancy Loss $\mathcal{L}_D$** To reduce the distribution discrepancy between different aspects, we introduce the aspect discrepancy loss (Gu et al., 2022c) to penalize the distance between distribution centers of each two aspects:

$$\mathcal{L}_D = \sum_{1 \le n_1 < n_2 \le N} \left\| \sum_{i \in S_{n_1}} \frac{z_i}{|S_{n_1}|} - \sum_{j \in S_{n_2}} \frac{z_j}{|S_{n_2}|} \right\|_2, \quad (3)$$

which calculates the Euclidean distance between two distribution centers. In practice, we use a batch-level approximation by taking the average representations of each aspect in each mini-batch as the estimated center and calculating the distances to

centers of other aspects. Minimizing $\mathcal{L}_D$ allows the model to reduce the discrepancy between different aspects, and helps to eliminate the mutual interference among them.

Totally, our learning objective is:

$$\mathcal{L} = w_1 \mathcal{L}_E + w_2 \mathcal{L}_C + w_3 \mathcal{L}_D. \quad (4)$$

We update parameters $\phi$, $\theta$ and $\{\pi_n\}$ for the encoder, decoder, and the classifier layers.

### 3.3 Formulating Joint Latent-Space EBMs

In order to satisfy the requirement of controlling multiple attributes simultaneously, we leverage the compositionality of EBMs and formulate the joint distribution for the latent representations and target attribute by incorporating any constraint(e.g., attribute classifiers) into the energy function $E(\cdot)$.

To begin with, we define the following joint distribution on both the latent representation $z$ and desired attributes $\boldsymbol{a}$ as:

$$p(z, \boldsymbol{a}) := p_{\text{prior}}(z)p(\boldsymbol{a}|z) = p_{\text{prior}}(z) \cdot e^{-E(\boldsymbol{a}|z)}/Z, \quad (5)$$

where $Z = \int e^{-E(\boldsymbol{a}|z)}dz$ is the normalization term, $p_{\text{prior}}(z)$ is the Gaussian prior distribution, and $p(\boldsymbol{a}|z)$ follows a Boltzmann distribution. In this work, We assume the target attributes are independent with each others. We then formulate $E(\boldsymbol{a}|z)$ as the energy-based models that can combine arbitrary attribute classifiers based on our needs:

$$E(\boldsymbol{a}|z) = \sum_{n=1}^{N} \lambda_n E_n\left(a_n^*|z\right). \quad (6)$$

$\lambda_n \in \mathbb{R}$ is the balanced weight to balance the performance among attributes from different aspects. The energy function $E_n(a_n^*|z)$ is defined as the negative log probability of target attribute $a_n^j$:

$$E_n(a_n^*|z) = -f_n(z)\left[a_n^j\right] + \log \sum_k \exp\left(f_n(z)\left[a_n^k\right]\right),$$

(7)

where $f_n(z)$ is the multi-class attribute classifier trained on the frozen latent space, and $f_n(z)[a_n^*]$ is the output unnormalized logits for attribute $a_n^*$. After the training of VAE, we fix the entire VAE encoder and map the input text with attribute annotations into the latent space, then ask the classifier to predict target attribute label given the latent vector. Training attribute classifiers $f_n(z)$ in the frozen low-dimensional latent space is efficient, which enables us to plug in different attribute classifiers to guide complex multi-aspect control.

### 3.4 Sampling from EBMs with ODE

After the acquisition of the joint distribution $p(z, \boldsymbol{a})$, we would like to draw latent representations $z$ given the target attribute values $\boldsymbol{a}$. To ensure high-quality and efficient sampling, we adopt a fast ODE-based sampler to draw samples from the energy based models.

Prior work (Song et al., 2021) shows that controllable generation $p(x|\boldsymbol{a})$ can be achieved by solving the following ordinary differential equation (ODE):

$$\mathrm{d}x = -\frac{1}{2}\beta(t)\left[x + \nabla_x \log p_t(x, \boldsymbol{a})\right]\mathrm{d}t, \quad (8)$$

where $\beta(t)$ is a time-variant diffusion coefficient that has the form $\beta(t) = \beta_{\min} + (\beta_{\max} - \beta_{\min})\,t$. $t$ is the timestep from $T$ to 0, and $p_t(x, \boldsymbol{a})$ denotes the join distribution of data and attribute at time $t$.

In our work, we adapt the ODE from Eq.(8) into the low-dimensional latent space, which gives:

$$\begin{aligned}\mathrm{d}z &= -\frac{1}{2}\beta(t)\left[z + \nabla_z \log p_t(z, \boldsymbol{a})\right]\mathrm{d}t \\ &= -\frac{1}{2}\beta(t)\left[z - \nabla_z E_t(\boldsymbol{a}|z) + \nabla_z \log p_t(z)\right]\mathrm{d}t.\end{aligned}$$

(9)

Note that $p_t(z) = \mathcal{N}(\mathbf{0}, I)$ is time-invariant for $t \in [0, T]$. Since the classifier $f_n(z)$ in Eq.(7) is fixed, $E_t(\boldsymbol{a}|z)$ is also time-invariant and we have $E_t(\boldsymbol{a}|z) = E(\boldsymbol{a}|z)$. The above ODE becomes:

$$\begin{aligned}\mathrm{d}z &= -\frac{1}{2}\beta(t)\left[z - \nabla_z E(\boldsymbol{a}|z) - \frac{1}{2}\nabla_z\|z\|_2^2\right]\mathrm{d}t \\ &= \frac{1}{2}\beta(t)\nabla_z E(\boldsymbol{a}|z)\,\mathrm{d}t \\ &= \frac{1}{2}\beta(t)\sum_n \nabla_z \lambda_n E_n(a_n^*|z)\,\mathrm{d}t.\end{aligned}$$

(10)

Now we can easily sample latent samples by

drawing $z(T) \sim \mathcal{N}(\mathbf{0}, I)$ and solving the Eq.(10) with a differential neural ODE solver[3] (Chen et al., 2018) to obtain $z(0)$. Then $z(0)$ is fed to the VAE decoder $p_\theta$ to produce target text sequences that possess multiple attributes simultaneously.

To narrow the inevitable gap between the prior distribution $p_{\mathrm{prior}}(z)$ and the learned VAE posterior $q_\phi(z|x)$ on $\mathcal{Z}$, following previous work (Li et al., 2020; Hu and Li, 2021; Liu et al., 2022), we fit a simple single-layer generative adversarial network (GAN) (Goodfellow et al., 2014), $p_{\mathrm{GAN}}(z)$, on the learned latent space and draw $z(T)$ from $p_{\mathrm{GAN}}(z)$. We study the impact of $p_{\mathrm{GAN}}$ in §4.5.

## 4 Experiments

In this section, we demonstrate the effectiveness of our proposed MacLaSa in the multi-aspect control setting through both automatic and human evaluations. Additionally, we provide further analysis and visualization on efficiency, and case studies.

### 4.1 Experimental Setups

**Datasets** We conduct experiments for controlling two aspects: sentiment and topic, simultaneously. We adopt the IMDb movie reviews (positive and negative) (Maas et al., 2011) for sentiment control and AGNews dataset (World, Sports, Business and Sci./Tech) (Zhang et al., 2015) for topic control. Following previous work (Qian et al., 2022; Gu et al., 2022c), we randomly sample 20k sentences from each dataset for each attribute to train our method. For evaluation, consistent with previous work (Dathathri et al., 2020; Krause et al., 2021; Yang and Klein, 2021; Liu et al., 2021; Gu et al., 2022a), we choose the same 15 attribute-unrelated prompts and ask the model to complete 50 sentences with the desired attributes starting with each prompt.

**MacLaSa Settings** For the proposed MacLaSa, we employ BERT-base and GPT-2 medium to initialize the encoder and decoder networks in VAE, respectively. The dimension of the latent space is 128. We also apply a cyclical schedule for KL weight and a KL thresholding scheme to alleviate the notorious KL vanishing issue (Bowman et al., 2016). During the training stage, we use the AdamW (Loshchilov and Hutter, 2017) optimizer with a learning rate of 8e-5. The number of training epochs is 50. We also randomly select 10k / 1k

---

[3] https://github.com/rtqichen/torchdiffeq

examples to train / validate attributes classifiers in the latent-space EBMs. In our experiments, $w_1$, $w_2$ and $w_3$ are set to 1. During the inference stage, we set $\beta_{\min} = 0.1$ and $\beta_{\max} = 20$ for the time-variant diffusion coefficient $\beta_t$. We also manually tune the weight $\lambda_n$ of different attributes to balance them. All experiments are conducted on a single NVIDIA V100 32GB GPU.

## 4.2 Baseline Models

We compare with three types of baseline models: (1) optimization-based methods: **PPLM** (Dathathri et al., 2020) back-propagates gradients of extra attribute classifiers to guide conditional generation at every decoding step. **DEXPERTS** (Liu et al., 2021) reweights the predictions of language models based on expert (and anti-expert) opinions for effective attribute control. **MaRCo** (Hallinan et al., 2023) achieves controllable generation using likelihoods under a expert LM and a anti-expert LM to find candidate words to mask and replace. **Mix&Match** (Mireshghallah et al., 2022) uses a Metropolis-Hastings sampling scheme to draw samplers from an energy-based model that combines multiple attribute discriminators. (2) Prefix-based methods: Contrastive Prefixes (abbreviated as **Contrastive**) (Qian et al., 2022) trains prefixes for each aspect while the combination of them can achieve multi-aspect control. We also compare with recent approaches that manipulate the latent space, including: **LatentOps** (Liu et al., 2022) performs composable text operations in the low-dimensional latent space, and **Distribution** (Gu et al., 2022c) searches for the intersection areas of multiple attribute distributions for generation.

## 4.3 Evaluation Metrics

**Automatic Evaluations** We adopt three automatic evaluations metrics to measure the performance on the two-aspect control task. **Correctness** evaluates the success rate of controlling the two aspects simultaneously. We finetune two RoBERTa-Large (Liu et al., 2019) discriminators on the IMDb dataset for sentiment aspect, and the AGNews dataset for topic aspect. We use the two attribute discriminators to compute the fraction of sentences that contain pre-specified attributes. **Perplexity (PPL)** is an automatic metric of text fluency. We feed generated test sentences to a GPT2-Large model and report the perplexity score. **Distinctness** (Li et al., 2016) is a n-gram-based metric for evaluating textual diversity, we report Distinct-1

and Distinct-2 in our paper.

**Human Evaluations** In addition to automatic evaluations, we conduct human evaluations to compare our method's performance with that of the baseline models. We enlist four annotators with high-level language skills to carry out the human evaluation. Annotators are instructed to assess attribute relevance, fluency, and diversity on a scale of 1-5, with 1 denoting "very low" and 5 representing "very high." Moreover, we direct the annotators not to consider linguistic quality when evaluating attribute alignment and vice versa. We randomly select 800 generated sentences (100 for each combination) and shuffle them for evaluation with each method. The scores are then averaged to derive the final human evaluation results.

## 4.4 Main Results

**Automatic Evaluations** We conduct experiments in the two-aspect control setting and compare our method with several strong baselines. The results of automatic evaluation are depicted in Table 1. We calculate the average correctness scores of eight attribute combinations as the final results for each method. We also report the standard deviations, which stand for the stability of models among different runs. Moreover, we assess the average inference time required to generate a single sentence for each method.

We note that existing baselines excel in individual evaluation metrics but struggle to concurrently achieve good controllability and superior linguistic quality, which is essential for multi-aspect control. PPLM and MaRCo can generate fluent sentences but fall short in attribute accuracy. In contrast, Mix&Match demonstrates strong attribute controllability, yet the text quality is subpar. Moreover, optimization-based methods, including PPLM and Mix&Match, exhibit severe slow inference speeds due to their complex iterations or searching in the high-dimensional text space. The Contrastive method attains a high correctness score in multi-aspect control by training separate continuous prefix vectors for each aspect. However, the mutual interference of different prefixes results in diminished text quality. LatentOps has average performance over baseline models. The Distribution method generates highly fluent texts with good attribute correctness scores but lacks textual diversity.

MacLaSa showcases a notable performance boost in average correctness scores, achieving an

| Method | Correctness (%) | Text Fluency | Diversity | | Efficiency |
|---|---|---|---|---|---|
| | Senti. & Topic Acc. ↑ | PPL ↓ | Distinct-1 ↑ | Distinct-2 ↑ | Time (s) ↓ |
| *optimization-based method* | | | | | |
| PPLM | $18.14 \pm 0.45$ | $25.59 \pm 1.09$ | 0.23 | 0.64 | 40.56 |
| DEXPERTS | $23.93 \pm 1.11$ | $38.70 \pm 2.51$ | 0.23 | 0.70 | 0.64 |
| MaRCo | $27.81 \pm 1.94$ | $18.87 \pm 1.85$ | 0.18 | 0.58 | 0.40 |
| Mix&Match | $50.17 \pm 2.07$ | $68.72 \pm 0.97$ | 0.36 | 0.84 | 164.60 |
| *prefix-based method* | | | | | |
| Contrastive | $53.02 \pm 1.52$ | $52.56 \pm 11.97$ | 0.22 | 0.71 | 0.59 |
| *method that manipulates latent space* | | | | | |
| LatentOps | $44.41 \pm 5.72$ | $26.11 \pm 1.46$ | 0.16 | 0.55 | 0.10 |
| Distribution | $49.79 \pm 1.99$ | $12.48 \pm 0.52$ | 0.08 | 0.28 | 0.04 |
| MacLaSa | $\mathbf{59.18 \pm 0.81}$ | $28.19 \pm 1.26$ | 0.16 | 0.60 | 0.10 |
| w/o $\mathcal{L}_C$ | $47.54 \pm 12.67$ | $27.91 \pm 1.10$ | 0.15 | 0.57 | 0.10 |
| w/o $\mathcal{L}_D$ | $51.18 \pm 3.90$ | $28.49 \pm 0.44$ | 0.18 | 0.62 | 0.10 |

Table 1: Automatic results on multi-aspect control. We average the correctness scores of eight combinations(two sentiment attributes × four topic attributes) as the final results for each method. Detailed results of each combination are listed in Appendix A.

| Method | Correctness↑ | Fluency↑ | Diversity↑ |
|---|---|---|---|
| PPLM | 1.96 | 2.67 | 2.54 |
| DEXPERTS | 1.98 | 2.38 | 1.88 |
| MaRCo | 2.08 | 2.78 | 2.65 |
| Mix&Match | 1.21 | 1.38 | 2.13 |
| Contrastive | 2.04 | 2.29 | 2.38 |
| LatentOps | 2.21 | 2.21 | 2.38 |
| Distribution | 2.67 | 2.67 | 2.63 |
| MacLaSa | **3.54** | **3.25** | **2.96** |

Table 2: Human evaluations on multi-aspect control.

| Method | Correctness (%) | Text Quality |
|---|---|---|
| | Sentiment & Topic ↑ | PPL ↓ |
| Random | 13.24 | 32.28 |
| LD | 26.93 | 6.70 |
| ODE (MacLaSa) | 58.22 | 26.86 |
| w/o $p_{\text{GAN}}$ | 37.82 | 36.93 |

Table 3: Automatic results of comparison of different samplers.

11.62% improvement compared to the strongest baseline. This result highlights our superiority in multi-aspect controllability. Additionally, MacLaSa displays good linguistic quality compared to previous method, emphasizing the benefits of learning a compact latent space. Our approach also exhibits substantial advantages in generation efficiency. Compared to the parameter-efficient prefix-based Contrastive method, our method demonstrates a remarkable 5.9× faster in inference speeds. In summary, MacLaSa surpasses existing baselines in attribute correctness and textual quality while keeping high inference speeds.

**Human Evaluations** The human evaluation results for the multi-aspect control task can be found in Table 2. The inter-annotator agreement is 0.32 in Fleiss' $\kappa$, indicating a fair agreement. Generally, the human judgment on attribute correctness aligns well with the results of the automatic evaluation. Our method excels in attribute control, achieving a correctness score of 3.54. Contrary to the automatic

evaluation results, annotators favor our approach as it delivers the highest text quality among the baselines. Overall, our model demonstrates superior performance in both attribute correctness and textual quality.

Both automatic and human evaluations demonstrate that our proposed MacLaSa outperforms other baseline models in terms of attribute correctness and linguistic quality, while maintaining a high inference speed.

### 4.5 Analysis

**Effects of VAE Losses** We conduct an ablation study to verify the effects of the classification loss $\mathcal{L}_C$ and aspect discrepancy loss $\mathcal{L}_D$. The results are shown in Table 1. Removing $\mathcal{L}_C$ causes the latent space to collapse completely. The correctness scores drop drastically as the model can hardly distinguish between representations of different attributes within the same aspect. Removing $\mathcal{L}_D$ degrades attribute correctness since we cannot alleviate domain gaps between different data sources. Interestingly, without $\mathcal{L}_D$, the distance between sam-

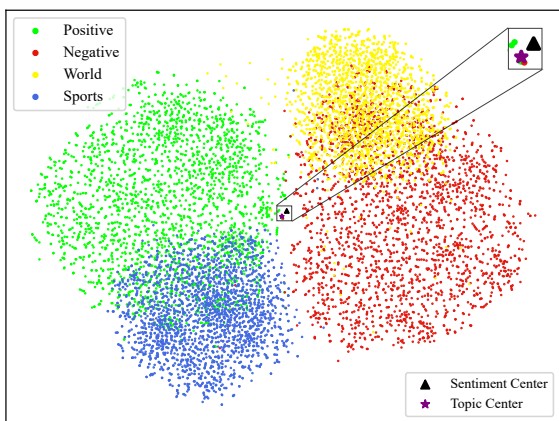

Figure 3: Projection of four attributes of two aspects from latent space via t-SNE.

| Model | Generated Sentences |
|---|---|
| Contrastive | Once upon a time, not so bad ASTRONAUT Tragic end-Mariners collapse to sweep from the cliff in the AL wild-goose division. |
| Contrastive | The country turns its tail WESTMINSTER is in many ways and SOUTHAMPTON FALL seems to have the same boring name of it all NBA names that is. |
| Contrastive | The president of the country not to be? Your unk will not like your unk, your unk says the singer, doesn. |
| Distribution | The last time was bad. The first time is bad. The third time is bad. And the fourth is worse. |
| Distribution | The horse is bad. The horse was bad in the first round of contest, showing a loss to rival South Korea after an earlier victory. |
| Distribution | The road is bad. The road was bad in the first round of competition, ending up with a record-breaking 30-year drought in the U.S. Open... |
| MacLaSa | The president of the country can't hang on to results. After Sunday's debacle in Philadelphia, there is little hope that Tedford University can maintain its ranking in the top 10 in the country. |
| MacLaSa | The horse was all wrong: Rossi Causes world championship leader Valentino Rossi suffered from an unusual form of ... |
| MacLaSa | The last time they clashed, they failed to meet expectations for this matchup to be made at the WNBA Finals and they have not been... |

Table 4: Example cases of generated sentences with attribute combination *negative* and *sports*. Red highlights sentiment-related contents. Blue highlights topic-related contents. Underlined are the input prompts.

ple points from different aspects increases, leading our model to generate sentences mapped from sparser regions. This results in a minor decrease in fluency while slightly increasing diversity.

**Effects of Samplers** To demonstrate the superiority of our ODE-based sampler, we compare it with other standard samplers. For fair comparison, we fix the parameters of VAE and choose different samplers for multi-aspect control text generation. We first implement a random sampler by directly drawing samples from the latent space using $p_{\text{GAN}}$(described in §3.4). We also compared it with a gradient-based sampler using Langevin Dynamics (Kumar et al., 2022; Qin et al., 2022). The automatic evaluation results are shown in Table 3. Random sampling directly from the latent space can only generate representations with single attributes, highlighting the necessity of using a specific sampler. While the LD-based sampler can generate high-quality sentences, it sacrifices attribute alignment, resulting in low attribute relevance. This may be because LD is sensitive and unrobust to hyperparameters (Nie et al., 2021). In contrast, our ODE-based sampler outperforms LD in terms of attribute alignment and textual diversity.

To investigate the impact of $p_{\text{GAN}}$, we conduct experiments by removing the GAN network and directly drawing latent representations from the standard Gaussian distribution $\mathcal{N}(\mathbf{0}, I)$. As shown in Table 3, without the GAN, our model cannot accurately estimate the attribute space, resulting in decreased attribute relevance and textual quality.

**Visualization of Latent Space** To provide an intuitive impression of the estimated latent space,

we use the t-SNE technique (Van der Maaten and Hinton, 2008) to visualize part of our estimated latent space with four attributes: *positive*, *negative*, *world* and *sports* in Figure 3. As shown, (1) attribute distributions within the same aspect are well separated due to the classification loss $\mathcal{L}_C$ that helps our model distinguish mutually exclusive attributes. (2) The distribution centers of sentiment and topic aspects are close to each other because we introduced $\mathcal{L}_D$ to penalize the distance between them to eliminate domain gaps, which helps generating high-quality multi-aspect sentences. We also notice that the combination of *negative-world* is tighter than that of *negative-sports* because *world* news often covers negative events such as war, disease, and famine. This observation aligns with our experimental results in Appendix A.

### 4.6 Case Study

To better understand the benefits of learning a compact latent space for generative models, we randomly present generated examples in Table 4. When generating sentences with the attribute combination *negative* and *sports*, the Contrastive method can generate attribute-related words like "*tragic*" and "*NBA*"; however, the semantic coherence of the sentences is insufficient. This observa-

tion is consistent with the results of both automatic and human evaluations (see § 4.4). One possible explanation is that the prefixes used for sentiment and topic control are trained independently, causing the two learned prefixes to exhibit different language habits and leading to incoherent expressions when combined for multi-aspect control. Conversely, the Distribution method can generate fluent sentences that display multiple attributes but struggles with varying expressions. For instance, Distribution tends to use the word "*bad*" to convey negative emotions, and its sentence structure is often repetitive, such as "*The [noun] was bad in the first round of*". Our proposed MacLaSa can generate numerous attribute-related content, such as "*there is little hope*" and "*world championship leader*", in a fluent manner. By minimizing the discrepancy between sentiment and topic representations in the latent space, we merge high-quality representations related to attribute information, resulting in more coherent expression.

## 5 Conclusion and Future Work

In this study, we introduce a novel method, namely MacLaSa, for multi-aspect controllable text generation that estimates a compact, low-dimensional latent space and employs a fast ODE-based sampler for efficient sampling. Our experiments on the two-aspect control task demonstrate the effectiveness and efficiency of our approach. Additionally, we carry out in-depth analytical experiments to emphasize the impact of each module and visualize the estimated latent space. In the future, we aim to expand our work by incorporating arbitrary attribute discriminators into the diffusion process using a plug-and-play approach. Furthermore, we plan to explore more powerful models to enhance the linguistic quality of generated sentences.

## Limitations

One of the limitations of the current MacLaSa approach is that when a new aspect or attribute is introduced, the entire VAE framework needs to be retrained to accommodate the unseen attributes. This retraining process can often be time-consuming and computationally expensive, posing a significant challenge in dynamic environments where new aspects may frequently emerge.

Moreover, due to the notorious KL vanishing issue, the training process of the VAE framework is not stable and requires a significant amount of skill and experience to address. The KL vanishing problem refers to the situation where, during the training process, the KL divergence term may approach zero. This can lead to a poorly constrained latent space, resulting in the model generating samples that lack diversity and are not representative of the true data distribution. To tackle this issue, we adopt several techniques, which are described in § 4.1.

## Ethics Statement

We honor and support the EMNLP code of Ethics. The paper focuses on controlled text generation, which aims to generate text with desired aspects. We recognize that controlled text generation may be misused to generate harmful texts, e.g., fake news. However, our method can also help to eliminate toxic information in pre-trained language models by introducing specific attribute classifiers. Overall, it is meaningful to continue research into this work based on predecessors. Besides, the datasets used in this paper are all from previously published work and do not involve privacy or ethical issues.

## Acknowledgements

This work was supported by the National Key R&D Program of China (2022YFB3103700, 2022YFB3103704), the National Natural Science Foundation of China (NSFC) under Grants No. 62276248, and the Youth Innovation Promotion Association CAS under Grants No. 2023111. Liang Pang is also supported by Beijing Academy of Artificial Intelligence (BAAI).

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

## A    Detailed Results of Multi-aspect Control

We exhibit the detailed results of eight combinations (two sentiment attributes × four topic attributes) on multi-aspect control in Table 5. We compare with three types of baselines: (1) optimization-based methods, PPLM and Mix&Match. (2) prefix-based method Contrastive, and (3) methods that manipulate the latent space, for example, LatentOps and the Distribution method. For automatic evaluation metrics, we fine-tune two RoBERTa-Large (Liu et al., 2019) discriminators to assess the attribute accuracy scores for both sentiment and topic aspects simultaneously. Perplexity (PPL) is employed to gauge the linguistic quality of the generated sentences. Additionally, we compute the Distinctness score to appraise the textual diversity, reporting Distinct-1 and Distinct-2 in our paper. We also report the standard deviations, which stand for the stability of models among different runs.

We observe that PPLM demonstrates strong controllability in specific combinations, such as the *Positive-Sci./Tech* pairing. However, the performance of each combination varies significantly, resulting in subpar average results. This phenomenon also exists for DEXPERTS and MaRCo. While Mix&Match and the Contrastive method excel at attribute alignment, their linguistic quality leaves much to be desired. We postulate that this is due to Mix&Match employing a Metropolis-Hastings sampling scheme for high-dimensional text space sampling, which is hindered by the discrete nature of text space and prevents smooth text generation. The Contrastive method posits that contrasting relationships between individual attributes within each aspect aid in training attribute controllers, but it neglects the differences between aspects, compromising overall textual quality. Regarding the two latent space manipulation methods, LatentOps exhibits moderate performance in both attribute relevance and textual quality, while the Distribution method generates fluent sentences with the desired attributes but lacks diversity.

Our method attains a remarkable average accuracy of 59.18% across the eight combinations, boasting a 11.62% improvement compared to the most powerful baseline and showcasing the exceptional controllability of our approach. Additionally, our technique excels in both linguistic quality and textual diversity. MacLaSa delivers well-rounded

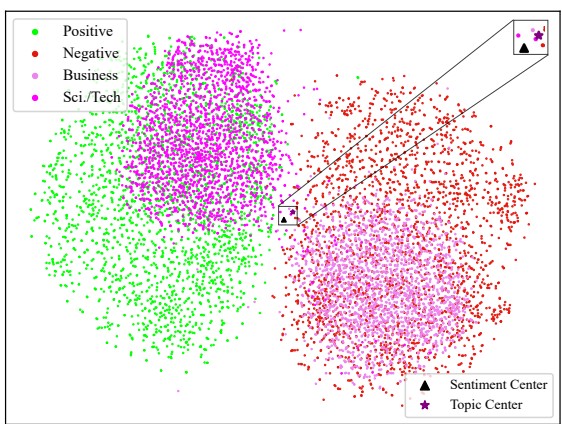

Figure 4: Projection of part of estimated attribute space with t-SNE.

performance concerning attribute alignment, linguistic quality, and diversity. We also evaluate the inference speed for sentence generation, with the results displayed in Table 1. The experimental findings indicate that MacLaSa maintains a high inference speed as well.

## B    Distribution of Attribute Space

In Figure 4, we employ the t-SNE technique to project hidden representations from four attributes into 2D for visualization: *positive*, *negative*, *business*, and *sci./tech*. This offers insight into a portion of the estimated latent space. We observe that, on one hand, the two sentiment attributes are distinctly separated due to the classification loss $\mathcal{L}_C$, which also applies to the topic aspects. Conversely, the distribution centers of the two aspects are situated closely together, as a result of the aspect discrepancy loss penalty $\mathcal{L}_D$. Overall, the observed attribute space distribution aligns with our expectations.

| Methods | Combination | Correctness (%) Senti. & Topic Acc. ↑ | Text Quality PPL ↓ | Diversity Distinct-1 ↑ | Distinct-2 ↑ |
|---|---|---|---|---|---|
| PPLM | Positive-World | 20.36 ± 1.69 | 25.47 ± 1.70 | 0.23 | 0.64 |
| | Positive-Sports | 16.53 ± 1.13 | 25.78 ± 1.30 | 0.23 | 0.63 |
| | Positive-Business | 25.24 ± 2.96 | 26.66 ± 1.26 | 0.24 | 0.64 |
| | Positive-Sci./Tech | 61.73 ± 0.66 | 25.06 ± 1.53 | 0.24 | 0.66 |
| | Negative-World | 3.87 ± 1.99 | 25.27 ± 1.23 | 0.23 | 0.64 |
| | Negative-Sports | 2.27 ± 0.57 | 25.96 ± 1.54 | 0.23 | 0.63 |
| | Negative-Business | 1.78 ± 1.26 | 26.11 ± 1.20 | 0.23 | 0.64 |
| | Negative-Sci./Tech | 13.29 ± 1.82 | 24.40 ± 1.11 | 0.24 | 0.66 |
| | *Average* | 18.14 ± 0.45 | 25.59 ± 1.09 | 0.23 | 0.64 |
| DEXPERTS | Positive-World | 34.22 ± 4.24 | 37.36 ± 3.46 | 0.24 | 0.72 |
| | Positive-Sports | 8.40 ± 2.66 | 37.36 ± 3.46 | 0.24 | 0.72 |
| | Positive-Business | 10.98 ± 1.67 | 37.36 ± 3.46 | 0.24 | 0.72 |
| | Positive-Sci./Tech | 45.02 ± 4.31 | 37.36 ± 3.46 | 0.24 | 0.72 |
| | Negative-World | 9.47 ± 2.68 | 40.03 ± 2.35 | 0.21 | 0.68 |
| | Negative-Sports | 8.17 ± 2.27 | 40.03 ± 2.35 | 0.21 | 0.68 |
| | Negative-Business | 10.98 ± 1.50 | 40.03 ± 2.35 | 0.21 | 0.68 |
| | Negative-Sci./Tech | 63.64 ± 8.73 | 40.03 ± 2.35 | 0.21 | 0.68 |
| | *Average* | 23.93 ± 1.11 | 38.70 ± 2.51 | 0.23 | 0.70 |
| MaRCo | Positive-World | 36.22 ± 8.04 | 17.13 ± 1.51 | 0.18 | 0.57 |
| | Positive-Sports | 37.11 ± 25.23 | 18.16 ± 1.47 | 0.17 | 0.55 |
| | Positive-Business | 38.89 ± 8.34 | 19.43 ± 2.13 | 0.19 | 0.59 |
| | Positive-Sci./Tech | 50.00 ± 5.21 | 17.91 ± 1.39 | 0.18 | 0.57 |
| | Negative-World | 8.22 ± 4.91 | 18.79 ± 1.88 | 0.19 | 0.59 |
| | Negative-Sports | 10.89 ± 0.38 | 19.94 ± 2.85 | 0.17 | 0.57 |
| | Negative-Business | 22.89 ± 20.07 | 20.51 ± 2.45 | 0.19 | 0.59 |
| | Negative-Sci./Tech | 18.22 ± 5.39 | 19.06 ± 1.91 | 0.18 | 0.59 |
| | *Average* | 27.81 ± 1.94 | 18.87 ± 1.85 | 0.18 | 0.58 |
| Mix&Match | Positive-World | 58.89 ± 0.83 | 61.27 ± 0.79 | 0.36 | 0.84 |
| | Positive-Sports | 58.89 ± 5.06 | 66.58 ± 2.52 | 0.35 | 0.84 |
| | Positive-Business | 39.78 ± 1.66 | 65.89 ± 1.77 | 0.35 | 0.84 |
| | Positive-Sci./Tech | 65.33 ± 2.49 | 69.07 ± 2.17 | 0.36 | 0.84 |
| | Negative-World | 41.55 ± 1.66 | 69.49 ± 1.14 | 0.35 | 0.84 |
| | Negative-Sports | 47.33 ± 8.13 | 72.72 ± 1.33 | 0.36 | 0.84 |
| | Negative-Business | 31.56 ± 5.15 | 71.61 ± 3.87 | 0.35 | 0.84 |
| | Negative-Sci./Tech | 58.00 ± 4.75 | 73.08 ± 2.06 | 0.37 | 0.84 |
| | *Average* | 50.17 ± 2.07 | 68.72 ± 0.97 | 0.36 | 0.84 |
| Contrastive | Positive-World | 67.87 ± 1.13 | 48.15 ± 15.74 | 0.23 | 0.72 |
| | Positive-Sports | 70.31 ± 5.55 | 52.36 ± 8.74 | 0.21 | 0.70 |
| | Positive-Business | 53.16 ± 5.00 | 56.13 ± 14.35 | 0.22 | 0.72 |
| | Positive-Sci./Tech | 51.96 ± 3.09 | 45.03 ± 12.27 | 0.23 | 0.71 |
| | Negative-World | 40.94 ± 4.26 | 51.27 ± 15.52 | 0.22 | 0.70 |
| | Negative-Sports | 40.71 ± 10.65 | 59.77 ± 8.87 | 0.21 | 0.71 |
| | Negative-Business | 48.84 ± 6.95 | 61.91 ± 15.14 | 0.20 | 0.70 |
| | Negative-Sci./Tech | 50.40 ± 3.95 | 45.86 ± 9.81 | 0.23 | 0.71 |
| | *Average* | 53.02 ± 1.52 | 52.56 ± 11.97 | 0.22 | 0.71 |
| LatentOps | Positive-World | 57.96 ± 5.07 | 24.79 ± 3.34 | 0.17 | 0.56 |
| | Positive-Sports | 63.47 ± 11.01 | 28.01 ± 1.80 | 0.16 | 0.55 |
| | Positive-Business | 61.73 ± 9.36 | 25.73 ± 1.84 | 0.14 | 0.52 |
| | Positive-Sci./Tech | 39.64 ± 22.07 | 26.49 ± 1.73 | 0.17 | 0.55 |
| | Negative-World | 34.62 ± 1.59 | 24.98 ± 1.56 | 0.16 | 0.55 |
| | Negative-Sports | 40.41 ± 9.72 | 25.14 ± 1.48 | 0.14 | 0.52 |
| | Negative-Business | 25.74 ± 2.41 | 27.30 ± 2.11 | 0.15 | 0.54 |
| | Negative-Sci./Tech | 31.56 ± 2.53 | 26.49 ± 0.99 | 0.16 | 0.57 |
| | *Average* | 44.41 ± 5.72 | 26.11 ± 1.46 | 0.16 | 0.55 |
| Distribution | Positive-World | 37.42 ± 4.38 | 13.34 ± 0.13 | 0.09 | 0.30 |
| | Positive-Sports | 71.60 ± 4.39 | 14.67 ± 0.53 | 0.09 | 0.29 |
| | Positive-Business | 72.80 ± 6.45 | 11.23 ± 1.00 | 0.07 | 0.25 |
| | Positive-Sci./Tech | 72.80 ± 11.07 | 12.41 ± 0.64 | 0.08 | 0.28 |
| | Negative-World | 46.80 ± 10.89 | 11.89 ± 1.12 | 0.07 | 0.28 |
| | Negative-Sports | 35.91 ± 7.84 | 12.99 ± 0.57 | 0.08 | 0.28 |
| | Negative-Business | 26.09 ± 5.60 | 11.03 ± 0.11 | 0.07 | 0.25 |
| | Negative-Sci./Tech | 34.86 ± 6.25 | 12.25 ± 0.93 | 0.08 | 0.27 |
| | *Average* | 49.79 ± 1.99 | 12.48 ± 0.52 | 0.08 | 0.28 |
| MacLaSa | Positive-World | 59.47 ± 6.66 | 26.26 ± 0.20 | 0.19 | 0.65 |
| | Positive-Sports | 87.93 ± 4.20 | 28.69 ± 1.78 | 0.16 | 0.57 |
| | Positive-Business | 82.87 ± 3.27 | 27.67 ± 1.55 | 0.15 | 0.57 |
| | Positive-Sci./Tech | 76.34 ± 0.46 | 28.77 ± 2.03 | 0.16 | 0.60 |
| | Negative-World | 56.54 ± 1.47 | 26.28 ± 1.26 | 0.16 | 0.59 |
| | Negative-Sports | 38.00 ± 2.67 | 32.23 ± 0.20 | 0.17 | 0.61 |
| | Negative-Business | 31.40 ± 4.07 | 29.06 ± 1.12 | 0.15 | 0.59 |
| | Negative-Sci./Tech | 44.74 ± 0.34 | 31.95 ± 0.48 | 0.17 | 0.62 |
| | *Average* | 59.18 ± 0.81 | 28.19 ± 1.26 | 0.16 | 0.60 |

Table 5: Detailed results of each combination on multi-aspect control.