# OpenReview forum: "MacLaSa: Multi-Aspect Controllable Text Generation via Efficient Sampling from Compact Latent Space"
_EMNLP/2023/Conference — EMNLP 2023 Findings_

### Official Review · Reviewer_YvKi · 2023-08-03

**Soundness:** 3

**Excitement:**

3: Ambivalent: It has merits (e.g., it reports state-of-the-art results, the idea is nice), but there are key weaknesses (e.g., it describes incremental work), and it can significantly benefit from another round of revision. However, I won't object to accepting it if my co-reviewers champion it.

**Paper Topic And Main Contributions:**

This paper proposes MacLaSa, an efficient framework for controllable text generation. MacLaSa first builds a latent space by VAE with classification and discrepancy regulation. Then, it builds an EBM on the latent space and samples the target latent vector with ODE. The experimental results show that MacLaSa performs well on multi-aspect controlling text generation.

**Reasons To Accept:**

1. The proposed method looks elegant and reasonable. It efficiently resolves the existing problem in multi-aspect controllable text generation.
2. The performance of MacLaSa is better than strong baselines.
3. The paper is well-writing.

**Reasons To Reject:**

1. The idea and overall framework of MacLaSa seems like a combination of LatentOPS and Distrbution. The sampling process of MacLaSa is very similar to LatentOPS. What's wore, Eq 5, Eq 7, and Eq 10 are all very similar to LatentOPS.
The difference between LatentOPS and Distribution, I think, is that the latent space of MacLaSa is more compact due to the usage of aspect discrepancy loss and classification Loss, however, which is proposed in LatentOPS. Therefore, I would like the author to discuss the difference between MacLaSa with LatentOPS and Distribution in detail.

**Reproducibility:**

4: Could mostly reproduce the results, but there may be some variation because of sample variance or minor variations in their interpretation of the protocol or method.

**Reviewer Confidence:**

4: Quite sure. I tried to check the important points carefully. It's unlikely, though conceivable, that I missed something that should affect my ratings.

---

> ### Author Rebuttal · Authors · 2023-08-27
>
> Thank you for your recognition and valuable suggestions. We appreciate your positive feedback on the elegant method, superior performance and good writing. We understand you concerns and would like to address it as follows:
>
> **Q1**: Difference between our method and prior research.
>
> **A1**: We understand you concern and we would like to clarify that: our model is specifically designed for multi-aspect control, which distinguishes it from LatentOps's single-aspect control.
>
> (1) We introduce suitable and usable solutions to the core challenges of multi-aspect control. To inject control signal during generation, we formulate the energy-based models in the low-dimensional continuous latent space estimated by a VAE, and plug in arbitrary classifiers to sample hidden representations with desired attributes based on our needs. To mitigate mutual interference between different controllers, we introduce the aspect discrepancy loss to penalize the distance between distribution centers of each two aspects. We also utilize an ODE-based sampler to facilitate fast sampling and ensure high efficiency.
>
> (2) The combination of these components is also non-trivial. To connect the VAE and ODEs, we fit a simple GAN on the latent space and sample from it. These samples are then fed into ODEs to obtain representations that bear multiple attributes. These representations are subsequently fed to the VAE decoder to produce target text sequences.
>
> (3) Through extensive experiments, we demonstrate that our method outperforms several strong baselines in terms of attribute alignment while keeping high efficiency and comparable linguistic quality.
>
> We believe that our approach provides a valuable contribution to the field by explicitly addressing the complexities of multi-aspect controllable text generation and achieving superior performance compared to existing methods. We are confident that our experimental results and the thorough analysis in our paper demonstrate the effectiveness and advantages of our proposed method.
>
> Thank your again for your feedback. We hope that our clarification addresses your concerns.

---

### Official Review · Reviewer_V3ko · 2023-08-03

**Soundness:** 4

**Excitement:**

3: Ambivalent: It has merits (e.g., it reports state-of-the-art results, the idea is nice), but there are key weaknesses (e.g., it describes incremental work), and it can significantly benefit from another round of revision. However, I won't object to accepting it if my co-reviewers champion it.

**Missing References:**

Controllable Unsupervised Text Attribute Transfer via Editing Entangled Latent Representation, NeurIPS 2019.

**Paper Topic And Main Contributions:**

This paper introduces a multi-aspect controllable text generation method that attempts eliminate the domain discrepancies by means of VAE and ODE latent samplers. The methods mentioned in the paper are a novel combination of well-known prior techniques. It is clearly stated in the paper how it differs from prior contributions and the work is well cited. The tasks used in the paper are well defined in the prior work.

**Reasons To Accept:**

The paper is mostly clearly written and has provided clear implementation instructions to reproduce results except for the human evaluation section.

The proposed ODE-based method demonstrates superior efficiency when compared with optimization and prefix-based methods.

The paper is well backed by experimental evidence and provides a  relatively comprehensive comparison with prior research. It represents a thorough and complete piece of work, showcasing the authors' meticulous evaluation of their approach.

**Reasons To Reject:**

The paper lacks sufficient experimental comparison and references concerning the optimization-based method, such as FGIM by Wang et al. , 2019.

Additionally, it is missing robustness analysis, considering the model's combination with ELBO loss.

Lack the time efficiency comparision with the optimization and prefix-based methods.

**Reproducibility:**

4: Could mostly reproduce the results, but there may be some variation because of sample variance or minor variations in their interpretation of the protocol or method.

**Reviewer Confidence:**

4: Quite sure. I tried to check the important points carefully. It's unlikely, though conceivable, that I missed something that should affect my ratings.

---

> ### Author Rebuttal · Authors · 2023-08-27
>
> Thank you for your recognition and valuable feedback. We appreciate your positive comments regarding our novel method, superior efficiency, comprehensive comparison and clear writing. We also understand your concerns and would like to address them as follows:
>
> **Q1**:  Experimental comparison with optimization-based methods, such as FGIM
>
> **A1**: We couldn't make a comparison with FGIM due to the fact that FGIM is primarily concerned with multi-aspect text style transfer, which is a different kind of generation task with our multi-aspect controllable text generation:
>
> - Multi-aspect text style transfer: transfer a source sentence with multiple attributes $a$ to another sentence with target multiple attributes $a'$.
>
> - Multi-aspect controllable text generation: generating a sentence with target multiple attributes $a'$ given a short prompt (e.g. "Once upon a time", not a complete sentence).
>
> Moreover, we would like to emphasize that our research focuses on a realistic scenario where sentences with multiple attribute labels are extremely scarce. In such a setting, the FGIM method is not applicable due to its reliance on input sentences with explicit attribute labels. The scarcity of sentences with multiple attribute labels is a common challenge in real-world applications, and our goal is to address this challenge by mapping sentences from different aspects into a joint latent space and incorporating specialized loss functions to minimize the distances between different attribute representations.
>
> Therefore, the methods like FGIM are not directly applicable to our task of multi-attribute control. This is the reason why we did not compare our work with FGIM or similar approaches in our previous work.
>
> We appreciate your understanding of the distinction between the text style transfer task and the limitations of applying FGIM to our specific problem. We will consider adding a brief discussion in the revised manuscript to address this point and further clarify the differences between our work and style transfer methods like FGIM.
>
> **Q2**: Robustness analysis.
>
> **A2**:  This point has been addressed in our original paper (see Table 1 on page 6). We have conducted experiments under different random seeds and reported the standard deviations of multiple results, which stand for the robustness of models under different runs. As can be seen, our method produces very stable results across different random seed settings, demonstrating the robustness and stability of our method.
>
> | Models               | Accuracy $\pm$ std |
> |:-------------------- |:------------------:|
> | PPLM                 | 18.14 $\pm$ 0.45   |
> | Mix&Match            | 50.17 $\pm$ 2.07   |
> | Contrastive Prefixes | 53.02 $\pm$ 1.52   |
> | LatentOps            | 44.41 $\pm$ 5.72   |
> | Distribution         | 49.79 $\pm$ 1.99   |
> | MacLaSa(Ours)        | 59.18 $\pm$ 0.81   |
>
> **Q3**: Time efficiency comparison
>
> **A3**: We would like to kindly draw your attention to the table 1 on page 6 in our original paper, where we have compared the execution efficiency of different methods in our original paper, which shows the average time to generate a complete sentence (in seconds).  Our method operates in a low-dimensional latent space and utilizes a fast sampler, which enables us to achieve higher efficiency compared with prior works.
>
> | Models               | Efficiency(s) |
> | -------------------- |:-------------:|
> | PPLM                 | 40.56         |
> | Mix&Match            | 164.60        |
> | Contrastive Prefixes | 0.59          |
> | LatentOps            | 0.10          |
> | Distribution         | 0.04          |
> | MacLaSa(Ours)        | 0.10          |
>
> **Q4**: Adding reference.
>
> **A4**: We appreciate your valuable suggestion. Since FGIM is primarily focused on the task of text style transfer, which is different from the objective of our study. We have discussed the detailed differences between our method with FGIM in A1 above. We will include the reference and the above discussion in our revised paper.
>
> In summary, we greatly appreciate your recognition of our strong performance and comprehensive comparison with prior research.  We have explained the differences between our approach and methods like FGIM, highlighting the unique nature of our controlled text generation task. We also draw your attention to the experimental results that you may have overlooked. We hope that this clarification addresses your concerns. If there are any further concerns, please let us know and we would be happy to address these concerns during the discussion phase.
>
> Thank you again for your time and effort. We look forward to your re-assessment of our paper.

---

### Official Review · Reviewer_2HjF · 2023-08-05

**Typos Grammar Style And Presentation Improvements:** The paper is written well and the fig…
**Soundness:** 3

**Excitement:**

3: Ambivalent: It has merits (e.g., it reports state-of-the-art results, the idea is nice), but there are key weaknesses (e.g., it describes incremental work), and it can significantly benefit from another round of revision. However, I won't object to accepting it if my co-reviewers champion it.

**Missing References:**

There are some relevant work about mapping sequences into latent space, doing unsupervised sentiment control, doing controllable text generation in general that are missing.

1) Detoxifying Text with MaRCo: Controllable Revision with Experts and Anti-Experts (Hallinan et al. 2022)

2) Extracting Latent Steering Vectors from Pretrained Language Models (Subramani et al. 2022)

3) Educating Text Autoencoders: Latent Representation Guidance via Denoising (Shen et al. 2020)

**Paper Topic And Main Contributions:**

This paper tackles the problem of doing multi-aspect controllable text generation, i.e. you have multiple attributes you want knobs to be able to turn when you're generating from a model. The authors introduce an approach MacLaSa that does this by estimating a latent space for multiple aspects together and then samples using a fast sampler. They evaluate performance on attribute relevance and textual quality.

**Questions For The Authors:**

1) Did you compare with DExperts (Liu et al. 2021) at all? There are also followup works from that group including Quark (Lu et al. 2022) and MaRCo (Hallinan et al. 2022) that are relevant baselines.

**Reasons To Accept:**

1) Multi-aspect control is an important problem. Ideally we want knobs for all these things and we want them to do this efficiently.

2) The performance numbers show gains over the included baselines.

3) The method from my reading of it, seems useful in any VAE-LM context.

**Reasons To Reject:**

1) There are a few baselines missing including DExperts (Liu et al 2021), Quark (Lu et al. 2022) and MaRCo (Hallinan et al. 2022) that could probably be integrated. These are much stronger numbers so I expect these methods to be much much stronger but potentially less efficient.

2) It'd be interesting to see more of an analysis of the latent space. There's a lot here that is missing. Some guiding points: Can you do interpolations in the latent space? Can you do vector arithmetic or other sort of geometric things? Do distances in the latent space make sense or reflect anything? What kinds of distances?

3) This seems hard to extend to decoder-only models like our current LLMs. This is a very minor point.


Thank you for the strong and careful rebuttal. It addressed many of these concerns:
- The missing baseline of DExperts has been added and shown to be poor (I imagine the comparison isn't super fair to their method, but DExperts is very specifically just good in the single controllable attribute setting). It's possible that you could mess with the baseline a little and tune it to be better, but for the sake of a rebuttal this is good. Retraining or reformulating Quark and MaRCo would be nice to see in a super fair comparison, but likely would take a couple of weeks to perhaps do well, so its beyond the scope of something me as a reviewer could reliably request in this format.

- The analysis of the latent space seems to indicate, perhaps further, that the method is doing what its supposed to and that the space is at least somewhat interpretable.

After the rebuttal, I raised my soundness score to a 3. I think there's definitely more that could be done, but I don't have any strong objections to it being accepted, so my excitement level remains a 3, but a stronger/higher 3 than it was before the rebuttal.

**Reproducibility:**

3: Could reproduce the results with some difficulty. The settings of parameters are underspecified or subjectively determined; the training/evaluation data are not widely available.

**Reviewer Confidence:**

3: Pretty sure, but there's a chance I missed something. Although I have a good feel for this area in general, I did not carefully check the paper's details, e.g., the math, experimental design, or novelty.

---

> ### Author Rebuttal · Authors · 2023-08-27
>
> Thank you for your constructive and valuable comments of our paper. We appreciate your positive feedback on our strong experimental results and wide potential applications in the context of multi-aspect controllable text generation. We understand your concerns and would like to address them as follows:
>
> **Q1**: Adding baselines (DExperts, Quark, and MaRCo, et al.)
>
> **A1**:  We appreciate your suggestions to include DExperts (Liu et al., 2021), Quark (Lu et al., 2022), and MaRCo (Hallinan et al., 2022) as additional baselines in our work. However, we would like to clarify that these papers mentioned above primarily focus on text detoxification experiments under the single-aspect control setting, which significantly differs from our objective of multi-aspect controllable text generation.
>
> However, to address your concern, we conduct additional experiments by adapting DExperts to the multi-aspect setting, as they had conducted sentiment control experiments (section 4 in their paper) that similar with our approach. We cannot compare our approach with Quark and MaRCo because they primarily focus on addressing the task of text detoxification under the single-aspect setting. Therefore, they are not applicable to our task of multi-aspect controllable generation.
>
> Our reproduced results of the DExperts method can be found in the new table below. It can be seen that DExperts performs poorly in the multi-aspect controllable generation setting. We speculate that this may be due to its neglect of the domain discrepancies among various aspects, leading to a significant performance deficiency. In contrast, our proposed method tackles this issue by mapping sentences from different aspects into a joint latent space and incorporating specialized loss functions to minimize the distances between different attribute representations. This enables us to effectively improve both control effectiveness and text quality in the multi-aspect controllable text generation task.
>
> We hope this clarifies the distinction between our work and the mentioned papers. We believe that our contribution lies in the development of a novel approach tailored for multi-aspect control, which is a more complex task compared to single-aspect control. We would be happy to provide further details and address any additional concerns you may have.
>
> | Models         | Accuracy $\uparrow$ | PPL $\downarrow$ | Dist-1 $\uparrow$ | Dist-2 $\uparrow$ | Efficiency $\downarrow$ |
> |:-------------- |:-------------------:|:----------------:|:-----------------:|:-----------------:|:-----------------------:|
> | DExperts       | 24.95               | 36.92            | 0.10              | 0.51              | 0.64                    |
> | MacLaSa (ours) | 59.18               | 28.19            | 0.16              | 0.60              | 0.10                    |
>
>
>
>
> **Q2**: More analysis of the latent space?
>
> **A2**: Thank you for your suggestions regarding the analysis of the latent space in our work.
>
> **Interpolation**: We conduct additional experiments to make it clearer. Specifically, we perform interpolation experiments in the latent space by sampling representations from different aspects and then interpolating between them to generate new latent representations, which are subsequently decoded into actual sentences. The results of interpolation in the learned latent space are listed in the table below.
>
> (1) The latent space we learned is a compact and well-behaved space with local continuity. We observed that the sentences generated from different interpolation combinations remained fluent, as indicated by their low Perplexity (PPL) scores. This suggests that the latent space captures meaningful representations that preserve the language coherence and quality of the generated sentences, reflecting the inherent structure and distribution of the data.
>
> (2) While linear interpolation in the latent space may seem intuitive for combining different attribute representations, it does not directly yield multi-aspect representations. This is because the latent space representations sampled from different attributes primarily retain the characteristics of single attributes. Therefore, a simple linear combination of these representations does not guarantee the generation of meaningful multi-aspect outputs. This highlight the need for specialized modules or techniques to facilitate the search or sampling of suitable representations that capture the desired multi-aspect representations, which is also supported by [1].
>
> To overcome this limitation, we leverage the composability of EBMs and formulate the joint distribution for the latent representations and target attribute by incorporating any constraint into the energy function $E(\cdot)$. Then we utilize a fast ODE-based sampler to draw samples from the EBMs and input them into the VAE decoder to generate the desired multi-aspect sequences.
>
> **Distance**: Regarding the distances in the latent space and their significance, in our study, we employed Euclidean distances to measure the distances between the distribution centers of different aspect. These distances are used to reduce the differences between different aspects, and we do not aim for their absolute values. Instead, we utilize a loss function to minimize these distances.
>
> It is important to note that these distances do not necessarily directly reflect semantic or grammatical differences. Rather, they serve as a metric tool to aid in finding appropriate representations in the latent space for achieving multi-attribute control. By optimizing the loss function, we gradually decrease the distances between different aspects, leading to improved multi-aspect control.
>
> We believe that our approach provides a valuable contribution to the field by explicitly addressing the challenges of multi-aspect controllable text generation and achieving superior performance compared to existing methods. We are confident that our experimental results and the thorough analysis in our paper demonstrate the effectiveness and advantages of our proposed method.
>
> We appreciate your interest in the latent space analysis and would be happy to provide any additional details or clarify any further questions you may have.
>
> | Interpolation | Accuracy $\uparrow$ | PPL $\downarrow$ | Dist-1 $\uparrow$ | Dist-2 $\uparrow$ |
> |:-------------:|:-------------------:|:----------------:|:-----------------:|:-----------------:|
> | 0.2           | 12.68               | 32.60             | 0.21              | 0.68              |
> | 0.4           | 12.72               | 32.47            | 0.21              | 0.68              |
> | 0.5           | 12.82               | 32.53            | 0.21              | 0.68              |
> | 0.6           | 13.97               | 31.74            | 0.21              | 0.67              |
> | 0.8           | 13.80               | 32.56            | 0.21              | 0.68              |
>
> Note: Interpolation means the Interpolation weight (ranging from 0 to 1) for the sentiment aspect and topic aspect:
>
> $$
> z_{\text{interpolation}} = z_{\text{senti}} * weight + z_{\text{topic}} * (1 - weight)
> $$
>
>
>
> **Q3**: Extending to decoder-only LLMs?
>
> **A3**: We want to clarify that our method is extendable to decoder-only models. In fact, our model operates within the framework of a Variational Autoencoder (VAE), which consists of an encoder and a decoder. The role of the decoder in this setup is to recover real sentences from the latent variables obtained through the encoding phrase. Therefore, the specific implementation of the decoder can utilize any pre-trained decoder-only language model. In our experiments, we chose to use the GPT-2 medium model for the decoder implementation to maintain a fair comparison with previous works. We believe this addresses your concern regarding the applicability of our method to decoder-only models like LLMs.
>
>
>
> **Q4**:  Adding baselines (DExperts)
>
> **A4**: We have addressed this concern in A1 above.
>
>
> **Q5**: Adding relevant work.
>
> **A5**: Thank you for your kind reminder of these relevant works. We agree with your point about the significance of single-aspect controllable text generation. We will include and discuss these relevant works in the revised version of our paper. We appreciate your guidance in helping us improve the completeness of our paper.
>
> In conclusion, we greatly appreciate your recognition of our strong performance and potential applications. We've added sufficient experiments according to your constructive reviews. We hope it will address your concerns and you can reconsider your assessment about the overall quality of our paper. We believe that our work offers an effective and efficient solution to the problem of multi-aspect controllable text generation, making a meaningful contribution to the community.
>
> Thank you once again for your recognition and contributions to our work.
>
> [1] Yuxuan Gu, Xiaocheng Feng, Sicheng Ma, Lingyuan Zhang, Heng Gong, and Bing Qin. 2022c. A distributional lens for multi-aspect controllable text generation. In Proceedings of the 2022 Conference on Empirical Methods in Natural Language Processing, EMNLP 2022, Abu Dhabi, United Arab Emirates, December 7-11, 2022.

---

### Meta-Review · Area_Chair_Rt8x · 2023-09-19

**Recommendation:** 3

**Metareview:**

The reviewers have reached on a consensus that the paper is well-written and acknowledged the effectiveness of the proposed method by looking at the gains compared to the baselines. The reviewers initially pointed out some missing baselines and comparison studies as well as some missing analysis of robustness, latent space characteristics, and efficiency. The authors were able to provide more in-depth studies and data points during the discussion period which mostly addressed the major concerns from the reviewers.

As a result, all the reviewers agree that the paper is good or strong at soundness (3, 4, 3). In terms of excitement, all the reviewers have reached on a consensus that it's ambivalent (3).

---

### Decision · Program_Chairs · 2023-10-07

**Decision:**

Accept-Findings

**Comment:**

The reviewers have reached on a consensus that the paper is well-written and acknowledged the effectiveness of the proposed method by looking at the gains compared to the baselines. The reviewers initially pointed out some missing baselines and comparison studies as well as some missing analysis of robustness, latent space characteristics, and efficiency. The authors were able to provide more in-depth studies and data points during the discussion period which mostly addressed the major concerns from the reviewers.

As a result, all the reviewers agree that the paper is good or strong at soundness (3, 4, 3). In terms of excitement, all the reviewers have reached on a consensus that it's ambivalent (3).